# The Availability of Culturally Preferred Fruits, Vegetables and Whole Grains in Corner Stores and Non-Traditional Food Stores

**DOI:** 10.3390/ijerph18095030

**Published:** 2021-05-10

**Authors:** Mary O. Hearst, Jade Yang, Samantha Friedrichsen, Kathleen Lenk, Caitlin Caspi, Melissa N. Laska

**Affiliations:** 1Public Health Department, St. Catherine University, Saint Paul, MN 55105, USA; jlyang102@stkate.edu; 2Professional Data Analysts, Minneapolis, MN 55418, USA; sfriedrichsen@pdastats.com; 3Division of Epidemiology and Community Health, University of Minnesota, Minneapolis, MN 55454, USA; lenk@umn.edu (K.L.); mnlaska@umn.edu (M.N.L.); 4Department of Allied Health Sciences, University of Connecticut, Hartford, CT 06103, USA; caitlin.caspi@uconn.edu

**Keywords:** food policy, cultural foods, food access

## Abstract

Chronic health inequities for communities of color is partially attributed to a lack of healthy preferred food access. This manuscript explores whether corner stores and non-traditional food stores stock fruits, vegetables and whole grain foods that the area cultural communities may prefer as part of complying with a local ordinance. This exploratory analysis identified corner and non-traditional food stores located in immigrant populations of color and African American neighborhoods as part of a larger study. Culturally preferred foods were identified from a list of food items in the parent (STORE) study and used to assess changes in availability. Stores did not have a great variety of culturally relevant foods pre- or post-ordinance, and overall findings show no significant changes over time and/or between ordinance and control community. Further interventions are needed to address cultural food availability in stores near communities of color.

## 1. Introduction

Limited access to culturally preferred foods may be an important contributing factor to poor dietary intake and chronic disease for populations of color [1,2]. The prevalence of hypertension (33%), overweight/obesity (69%) and diabetes (12.4%) among adults in the United States (U.S.) remains high [3,4]; however, there is a disproportionate burden on immigrant populations of color and African American populations [1,5]. For example, African Americans (49.6%) have the highest age-adjusted prevalence of obesity, followed by Latinxs (44.8%), non-Latinx whites (42.2%) and non-Latinx Asians (17.4%) [6]. The prevalence of diabetes in the U.S. follows a similar pattern [7]. Findings from the 2010–2016 U.S. National Health Interview Survey indicate the prevalence of diagnosed hypertension, overweight/obesity and diabetes among individuals born in Mexico/Central America/Caribbean are 27.7% for hypertension, 70.7% for overweight/obesity, 11.6% for diabetes, and are similarly high for those born in Africa and Southeast Asia [5].

Previous research indicates that healthy consumption of fruits and vegetables greatly contributes to chronic disease prevention and management, but fruit and vegetable intakes fall below recommended levels particularly for immigrant populations of color and African Americans [1]. Numerous structural barriers contribute to the limited consumption of healthy foods among these groups, some of which are specific to immigrant groups, and others of which are more broadly rooted in longstanding economic injustice and social oppression of racialized groups in the U.S. [8,9]. Residential segregation and prolonged economic disinvestment in many predominantly African American communities has led to neighborhood food access disparities. This landscape includes limited access to full-service supermarkets and fresh produce and greater access to convenience and non-traditional stores in these neighborhoods compared to predominantly white neighborhoods [10,11]. Shopping at or living near large supermarkets or grocery stores has been associated with higher consumption of fruits and vegetables, while close proximity to convenience and small stores has been associated with lower intake. Recent immigrants may face additional barriers to healthy eating, including unfamiliarity of new foods and exposure to new convenience foods [12]. Finding healthy and familiar foods for the purpose of maintaining cultural traditions and affirming group identity can be difficult [11]. In general, dietary acculturation (the transition from traditional diets to typical American styles of eating) may contribute to the increasing risk of chronic disease among immigrants [13,14,15,16,17,18,19].

Industry organizations have increasingly encouraged convenience store and small store owners to reach out to the untapped growth market, specifically African American, Asian and Latinx demographics [20]. Retailers are also being encouraged to better align themselves with consumer needs and motivations [20] and provide products in their stores that are connected to their customer heritage [21]. Growing demand for ethnic foods is particularly salient for the Latinx population [22] and expanding availability of ethnic foods across the entire customer base [23].

Local policy is a possible tool to improve place-based food access disparities that have contributed to racial health disparities [1]. The Minneapolis, Minnesota, USA, City Council passed a Staple Foods Ordinance in 2014, the purpose of which was to “ensure that everyone has access to healthy foods no matter where they shop” [24]. The ordinance required a minimum stocking requirement for nutrient-rich foods in 10 food categories, including quality indicators for perishable items. Licensed retailers were contacted by the Minneapolis Health Department, which provided compliance assessments and education on requirements. The tools/training provided were designed with specific attention to cultural foods in particular, but they were intended to be relevant to all types of stores and to be flexible with store needs. Enforcement began in 2016, when Minneapolis Health Department employees conducted inspections of the licensed retailers. Consequences for non-compliance ranged from warning letters to fines. Detailed history, ordinance revision and compliance can be found in [25]. In the context of local efforts to meet community needs for healthy food and recommendations by industry to align products with the local customer base, this paper explores whether the corner stores and non-traditional stores met the requirements by stocking preferred foods based on area demographics. This paper focuses on Black/African American, Asian, Latinx and East African residents. The ordinance intentionally created categories broad enough that minimum standards could be met in numerous ways. The hypothesis for this analysis was that corner and non-traditional food stores located in immigrant populations of color and African American neighborhoods would be more likely to provide culturally preferred foods. It was subsequently hypothesized that the stores would increase the availability of culturally preferred foods to meet both ordinance requirements and the market demands of the community in which the store is located. In order to determine if the hypothesized change in availability is due to implementation of the ordinance rather than a larger industry shift in what is available in stores, change will be tested against a control community that did not implement an ordinance.

## 2. Materials and Methods

The present study was a secondary data analysis with data collected in the STORE (STaple foods ORdinance Evaluation) study. The STORE study tested compliance to the ordinance among convenience stores and small, non-traditional food stores (e.g., gas food marts, dollar stores), as well as impact on customer purchasing and home food environments. Retailer compliance was assessed at four different time points during the implementation and follow-up period. Data were collected pre-policy (July–December 2014, hereafter called time 1), during an implementation-only phase (no enforcement; September–October 2015, hereafter called time 2), at the initiation of enforcement (May–July 2016, hereafter time 3) and after continued monitoring (August–December 2017, hereafter time 4). The STORE study compared changes in food availability at stores between neighboring cities, Minneapolis (ordinance implementation) and Saint Paul (no ordinance). Ninety stores per city were randomly selected from a government list of stores with grocery licenses. Exclusion criteria included supermarkets; WIC-authorized retailers, as they were likely already compliant; invalid licensing addresses or exemption from the ordinance due to size; specialization or other exemptions (*n* = 255). Following field visits, verification and consent, 159 stores actively consented to participate in the study at one or more of the four data collection points [25].

### 2.1. Store Assessments

Trained staff assessed the availability and price of 69 food items using a modified instrument from the Rudd Center for Food Policy and Obesity [26]. The instrument lists items in specific package sizes for which availability, price and quality (e.g., fruits and vegetables) is recorded. The adapted instrument can be found here: https://conservancy.umn.edu/handle/11299/20378, [25]. The parent study evaluated the change in the availability and price of all 69 items in the Minneapolis stores and a comparison sample in Saint Paul, a comparable neighboring city. Other store characteristics were collected, including store ownership (independent versus corporate), EBT/SNAP authorization and store location [27]. Store ownership may be an important determinant of manager decisions around stocking healthy food.

For the present exploratory analysis, culturally preferred foods that were included on the STORE assessment were identified based on data from The Food Group, an equity-focused local food bank that compiled the food list through client requests and key informant interviews in its process of creating a cultural equity toolkit [10]. This resource was selected, because it provided a local perspective of high-demand foods informed by community leaders, and it was supplemented by published literature that included studies in other areas of the U.S. Notably, if a food appeared on the culturally preferred list, it did not imply that the food is uniquely appealing to or consumed by certain groups, only that the food may have demand, serve as a household staple, or be more commonly part of the life experience among certain groups.

There were 42 unique food items determined to be culturally preferred across the focus populations. Black/African American-preferred foods were identified as bananas, peaches, blackberries, blueberries, raspberries, tomatoes, collard greens, corn, kale, okra, turnip, yam, dry lentils/peas, cornmeal, fufu and millet [1,28]. Many East African (a subpopulation of the Black/African American and highly prevalent in Minnesota [29]) consumption patterns and preferences overlap with Black/African American, including dry lentils/peas, tomatoes, millet and yams [1]. Additional cultural foods for East African communities were identified as dry beans, corn and teff. Latinx preferred foods were identified as bananas, pineapples, avocados, guavas, limes, mangos, papaya, tomatoes, acorn squash, peppers, plantains, yellow squash, zucchini squash, dry beans, dry lentils/peas, whole wheat tortillas, white corn tortillas, white flour tortillas and cornmeal/masa [1,10]. Commonly consumed Asian foods were identified as tofu, bananas, oranges, peaches, limes, pears, broccoli, green/red cabbage, bok choy and eggplant [30,31].

The 5-year American Community Survey estimates (ACS, 2009–2014) [32] were used to determine community demographics. A community was defined as the census tract where each store was located. We identified four types of communities of color for this paper, where 20% of the census tract population was either Black/African American, Latinx, Asian or East African [33]. East African communities were determined by a 20% or greater Black/African American population with an additional language spoken at home. Once these communities were indicated, local knowledge was used to confirm the identification of high presence of East Africans living is those areas.

### 2.2. Analysis Methods

The analysis included stores (corner stores, gas marts and dollar stores) located in one of the four types of communities of color described above and that were assessed at both the pre-ordinance time point (time 1) and three years following implementation (time 4).

Store characteristics were summarized overall for Minneapolis (ordinance) and Saint Paul (control) stores and by the four types of communities of color for Minneapolis stores using descriptive statistics. Two-sample *t*-tests or chi-square tests (or Fisher’s exact tests where any cell count <5) were used to compare the community and store characteristics between the Saint Paul and Minneapolis stores. Statistical tests were not run to compare stores in each community of color, because they were not mutually exclusive.

Descriptive statistics (frequencies and proportions) were calculated for the availability of the cultural foods at time 1 and time 4 (for foods present in at least one store in Minneapolis at either time point), overall and stratified by the community of color. McNemar’s exact tests were computed to test for statistically significant changes in the availability of each cultural food (as well as any cultural food) from time 1 to time 4. A paired *t*-test was computed to test for a change in the average number of culturally available foods from time 1 to time 4 within each community of color. Since the number of culturally available foods was not completely normally distributed, in addition to paired t-tests, we ran nonparametric Wilcoxon signed-rank tests and compared the findings. The interpretation of the analyses did not differ with parametric or non-parametric testing. Therefore, parametric test results are reported.

To test the change in the availability of at least one cultural food in stores, we used generalized linear mixed models with a random intercept for a store, specifying a binary distribution and an identity link. The outcome was the availability of at least one cultural food (yes/no) within each community of color (where model converged and *n* ≥ 20) and for the full sample. Models were run unadjusted as well as after adjusting for store ownership type (independent vs. corporate). A second set of regression models were computed with Minneapolis stores only, overall and stratified by each community of color (where model converged and *n* ≥ 10), testing for a change in the availability of any cultural food from pre-ordinance to 12-months post-ordinance enforcement, accounting for store ownership type (independent vs. corporate).

Regression models were only computed when the number of stores was at least 10 per city and the total number of stores in the model was at least 20. Due to a small number of stores (*n* = 2) in Asian communities in Minneapolis, computing regression models were not possible for Asian communities. Similarly, due to the small number stores in Latinx (*n* = 3) and East African (*n* = 5) communities in St. Paul, models for these communities of color were limited to Minneapolis only. SAS v.9.4 (SAS Institute Inc., Cary, NC, USA) was used for analysis. *p*-values < 0.05 were considered statistically significant, and 95% confidence intervals were provided where appropriate.

## 3. Results

The final analytic sample size was 60 stores, with 31 in Minneapolis and 29 in St. Paul. The final neighborhood-type store counts (for both cities combined) are: Black/African American (*n* = 39), East African (*n* = 19), Asian (*n* = 26), Latinx (*n* = 16). In Minneapolis, 26 identified stores in the sample were in Black/African American communities, 14 in East African communities and 13 in Latinx communities. Store characteristics, including size (number of aisles and cash registers), EBT/SNAP acceptance and ownership type were similar between cities. See Table 1.

In Minneapolis, 80.7% of stores had at least one culturally relevant food available pre-ordinance, compared to 90.3% post-ordinance, a difference that was not statistically different. There were few cultural foods available in Minneapolis before or after the ordinance, and none of the changes over time were statistically significant (Table 2). There were no notable changes in items available for all three cultural communities. For Asian cultural foods at stores in Asian neighborhoods (*n* = 2), one store did not have any cultural foods pre- and post- (bananas, oranges, and limes pre-; and bananas and oranges post-; data not shown in tables). The following foods were not available anywhere pre- or post-: Black/African American: blackberries, blueberries, raspberries, chard, collard greens, kale, yams, rutabaga, peaches, beets, okra; East African: yams, corn, millet; Asian: tofu, peaches, pears, broccoli, green cabbage, eggplant, red cabbage; Latinx: avocados, acorn squash, rutabaga, cornmeal/masa (data not shown in table).

The results for the models assessing changes in availability in Minneapolis and St. Paul over time for all cultural communities combined, as well as specifically for Black/African American communities, are presented in Table 3. The presence of any type of cultural food at stores in Minneapolis increased 9.7% from pre- to post-; however, there was also 3.4% increase for Saint Paul. After adjusting for store ownership type, the change for Minneapolis was 6.1%, while the change for St. Paul was 4.0%; none of these changes were statistically significant. Among stores in Minneapolis, the availability of any cultural food item did not change for independently owned stores, but it increased from pre- to post- for corporate-owned stores by 23% (not statistically significantly different within or across cities). Among stores within Black/African American communities, there were no statistically significant changes from pre- to post-, though patterns suggest a slight increase in cultural food availability in these stores in Minneapolis and a slight decrease in St. Paul.

## 4. Discussion

Our exploratory analysis suggests convenience stores and small, non-traditional food stores in census tracts in communities of color had very limited varieties of culturally preferred food items. The Minneapolis Staple Foods Ordinance was implemented to improve quantity and variety of healthy and staple foods in convenience and other small, non-traditional food stores, which are disproportionately more common than supermarkets in communities of color [11]. Generally, these results do not provide evidence that convenience and other small, non-traditional food stores chose to meet the ordinance minimum standards by focusing on culturally preferred food of the cultural communities in the surrounding area. In fact, there were very few of the culturally specific foods assessed that were available before or after the ordinance. The presence of any culturally specific food at stores in Minneapolis increased slightly from pre- to post-ordinance; however, there was also a slight increase for St. Paul, and these changes were not significant. This is consistent with the primary study findings for healthy food availability in general across all stores, specifically that, while healthy food availability increased in Minneapolis, there was also an increase in St. Paul [25]. Therefore, the parent study highlighted a possible industry trend toward providing healthier foods in general [25], and this study found a similar industry trend for culturally preferred foods, although insignificant. Consider that the foods that were available tended to be universal or common in a U.S. diet (e.g., tomatoes, bananas, corn, berries) [34].

Increasing access to large supermarkets and grocery stores may help with preventing conditions related to diet. However, supermarkets are inequitably distributed, leaving small food stores and convenience stores as a crucial source for food in many communities, particularly communities of color. The lack of access to full-service food stores is rooted in systemic racism and compounded by a larger system of disinvestment of communities of color that includes economic isolation and a lack of urban infrastructure and that ultimately affects exposure to chronic disease in these populations [8,9].

Dietary changes are contextual. For Black/African Americans in particular, the history of slavery heavily influenced identified culturally preferred foods [35]. When slave ships kidnapped Africans for slavery, ships brought some crops from West Africa (e.g., watermelon, okra, peanuts) to the U.S. for the slaves to farm and eat. Other traditional foods were not available to the slaves, such as African yams, and they adopted the sweet potato as a similar food item [35,36]. The roots of “Soul Food” are a product of using available food ingredients to maintain elements of original West African meals [36]. A similar yet more dramatic dietary pattern shift due to contextual conditions is seen among Native Americans [37], not discussed here.

One potentially interesting finding is that, among stores in Minneapolis, the availability of any cultural food item did not change for independently owned stores, but it increased from pre- to post- for corporate-owned stores by 23% (though this was not a statistically significant change). This may be related to the fact that the corporate stores in Minneapolis started with a lower percentage availability of cultural foods than the independent stores at baseline. Targeted marketing of unhealthy food to disadvantaged groups, including African American adults and youth, has a long been a practice of the food industry [38,39]. An opportunity exists, supported by industry and the public health literature, alike, for convenience stores to provide more variety and promotion of healthy and culturally relevant foods specific to their communities, which could serve to both increase revenue and address health disparities among immigrant populations of color in particular [21,22,23]. It is possible that corporate stores are more likely to follow the industry literature and be more able to respond to trends, as opposed to smaller, independent storeowners. Corporate stores also likely have fewer challenges in food distribution, particularly related to perishable foods, especially when foods are seasonal. It is unknown how the stores in this study accessed seasonal produce. Independent stores take on more risk than corporate stores as they change their stock and identify distributors that will deliver perishable foods in small quantities and frequently. However, these hypotheses would not explain the difference between the intervention and control condition, and this requires further analysis. One potential intervention could be finding ways to inform smaller, independent storeowners of market trends and the potential value in meeting the cultural preferences of the community demographics. Support could be offered for independent stores to model the costs and benefits associated perishable food options and seek economies of scale with other stores. Further information is also needed to determine what makes a store welcoming to certain cultural communities, such as a minimum threshold of cultural foods available and/or features of the store environment.

## 5. Limitations

There are several limitations of this study to consider. First, the ordinance and the data collection were not originally intended to evaluate the impact on culturally available foods. There are also challenges to using any culturally preferred food list; while the purpose of such a list was to identify potentially high-demand foods, it risks oversimplifying or generalizing complex cultural food preferences. The decision to use 20% as the indicator of a culturally influenced census tract was built on the data surrounding “white flight”—meaning that, once a white neighborhood reaches 20% persons of color, there is a dramatic rise in white residents moving away from the neighborhood [33]—yet may not have adequate population size to warrant specific food stocking due to limited purchasing power. Finally, while this study was meant to examine the local food environment in neighborhoods with particular demographics, it must be noted that individual shopping patterns align imperfectly with census tracts and to proximity to stores. Individuals make decisions about where to shop based on an array of social, economic and geographic factors and routinely shop at stores outside their neighborhood [40,41,42]. Despite these limitations, the original study design was rigorous, and these exploratory analysis results are consistent with the main outcomes study [25]. The Minneapolis Staple Foods Ordinance is the first and one of the only local policies of its kind, and findings from its evaluation provide unique insights for future efforts. In general, few studies have employed a cohort design in examining retail food environments within small food stores, and natural experiments in this area are also lacking.

## 6. Conclusions

Black/African American and communities of color have a continued disadvantage in accessing foods where they live due to systemic racism. There is much work to be done to build and sustain food justices in the U.S., even in the place of a well-intentioned policy. Further research is needed to fully understand the relationship between store owners and their community; the demand for fruits, vegetables and whole grain from these sources; and the stability of “culturally preferred” foods through generations. While there is an increase in fruits, vegetables and whole grain food items more available, more work is needed to achieve health equity.

## Figures and Tables

**Table 1 ijerph-18-05030-t001:** Store and neighborhood characteristics at baseline (pre-ordinance, 2014).

*n* (%) or Mean (sd)	Minneapolis Stores by Neighborhood (≥20% Based on Census Tract Demographics) ^1^					
	Minneapolis	Saint Paul	*p*-Value	Black/African American	East African	Hispanic
*N*	31	29		26	14	13
Store type, *n* (%)			0.978			
Corner store, convenience store, or small grocery	14 (45.2)	11 (37.9)		12 (46.2)	8 (57.1)	6 (46.2)
Food–gas mart	10 (32.3)	11 (37.9)		8 (30.8)	3 (21.4)	5 (38.5)
Dollar store	3 (9.7)	3 (10.3)		3 (11.5)	1 (7.1)	1 (7.7)
Pharmacy	4 (12.9)	4 (13.8)		3 (11.5)	2 (14.3)	1 (7.7)
Number of store aisles, *n* (%)						
0–4	11 (36.7)	11 (40.7)	0.947	10 (40.0)	7 (50.0)	6 (46.2)
5–8	11 (36.7)	9 (33.3)		8 (32.0)	4 (28.6)	3 (23.1)
9+	8 (26.7)	7 (25.9)		7 (25.0)	3 (21.4)	4 (30.8)
Number of cash registers, *n* (%)			0.740			
1	14 (46.7)	10 (37.0)		14 (56.0)	9 (64.3)	4 (30.8)
2–3	11 (36.7)	11 (40.7)		7 (28.0)	2 (14.3)	7 (53.9)
4+	5 (16.7)	6 (22.2)		4 (16.0)	3 (21.4)	2 (15.4)
EBT/SNAP accepted, *n* (% yes)	30 (96.8)	25 (96.2)	1.000	25 (96.2)	13 (92.9)	13 (100.0)
Ownership type, n (% independent)	18 (58.1)	12 (41.4)	0.301	16 (61.5)	11 (78.6)	8 (61.5)
Neighborhood demographics, mean % (sd)						
Poverty (below 185% poverty level)	52.7 (15.3)	48.1 (14.5)	0.234	56.2 (13.5)	58.0 (11)	50.9 (18.7)
Hispanic	16.5 (12.8)	12.1 (6.3)	0.100	14.9 (13.4)	15.6 (12.9)	29.4 (8.8)
non-Hispanic White	37.5 (19.0)	36.1 (14.4)	0.764	33.6 (17.6)	41.6 (14.5)	36.6 (22.6)
non-Hispanic Black/African American	31.6 (15.9)	20.1 (15.4)	0.006	35.9 (13.1)	32.7 (9.6)	19.6 (12.6)
non-Hispanic American Indian/Alaska Native	2.2 (3.3)	1.0 (1.1)	0.051	2.5 (3.5)	3.0 (4.0)	2.5 (3.1)
non-Hispanic Asian	7.6 (8.9)	26.9 (8.9)	<0.001	8.5 (9.5)	3.6 (2.8)	7.2 (9.0)
non-Hispanic Native Hawaiian or Pacific Islander	0.0 (0.1)	0.1 (0.3)	0.236	0.0 (0.1)	0.1 (0.1)	0.0 (0.0)
non-Hispanic other	0.2 (0.7)	0.1 (0.2)	0.330	0.3 (0.8)	0.3 (1.0)	0.0 (0.1)
More than one race	4.9 (2.7)	4.1 (2.2)	0.205	4.9 (2.8)	3.6 (1.6)	5.5 (2.4)
Stores in Black/African American neighborhood ^2^, *n* (%)	26 (83.9)	13 (44.8)		26 (100.0)	14 (100.0)	8 (61.5)
Stores in East African neighborhood ^2^, *n* (%)	14 (45.2)	5 (17.2)		14 (53.9)	14 (100.0)	5 (38.5)
Stores in Asian neighborhood ^2^, *n* (%)	2 (6.5)	24 (82.8)		2 (7.7)	0 (0.0)	1 (7.7)
Stores in Hispanic neighborhood ^2^, *n* (%)	13 (41.9)	3 (10.3)		8 (30.8)	5 (35.7)	13 (100.0)

Notes. *p*-values from chi-square tests, Fisher’s exact tests or two-sample *t*-tests; ^1^ Not reporting descriptive statistics for the *n* = 2 Asian stores because of small sample size; ^2^ Not mutually exclusive and represents the study sample only.

**Table 2 ijerph-18-05030-t002:** Availability of specific cultural foods at Minneapolis stores by race/ethnicity neighborhood grouping pre- and 12-months post-ordinance enforcement—Unadjusted results.

*n* (%) or Mean (sd)	Pre-	Post-	*p*-Value
All stores: any cultural food (*n* = 31)	25 (80.7%)	28 (90.3%)	0.180
Black/African American (*n* = 26 stores)			
Any cultural food	18 (69.2%)	21 (80.8%)	0.180
Bananas	16 (61.5%)	13 (50.0%)	0.257
Watermelon	1 (3.9%)	0 (0%)	
Tomatoes	4 (15.4%)	8 (30.8%)	0.157
Corn	0 (0%)	1 (3.9%)	
Chili pepper	4 (15.4%)	2 (7.7%)	0.157
Turnips	1 (3.85%)	0 (0%)	
Lentils	10 (38.5%)	16 (61.5%)	0.058
Masa/cornmeal	3 (11.5%)	1 (3.9%)	0.157
Millet	0 (0%)	1 (3.9%)	
Avg. number of foods available	1.50 (1.39)	1.62 (1.24)	0.640
East African (*n* = 14 stores)			
Any cultural food	8 (57.1%)	11 (78.6%)	0.180
Tomatoes	2 (14.3%)	4 (28.6%)	0.157
Dry beans	8 (61.5%)	10 (76.9%)	0.317
Lentils	7 (50.0%)	9 (64.3%)	0.414
Avg. number of foods available	1.21 (1.19)	1.64 (1.15)	0.189
Hispanic (*n* = 13 stores)			
Any cultural food	12 (92.3%)	13 (100%)	
Bananas	8 (61.5%)	9 (69.2%)	
Pineapples	1 (7.7%)	1 (7.7%)	
Avocados	1 (7.7%)	1 (7.7%)	
Guavas	1 (7.7%)	1 (7.7%)	
Limes	4 (30.8%)	5 (38.5%)	0.564
Mangoes	2 (15.4%)	0 (0%)	0.500
Papayas	0 (0.0%)	1 (7.7%)	
Tomatoes	2 (15.4%)	3 (23.1%)	0.564
Beets	1 (7.7%)	1 (7.7%)	
Red bell peppers	1 (7.7%)	1 (7.7%)	
Plantains	1 (7.7%)	1 (7.7%)	
Zucchinis	1 (7.7%)	1 (7.7%)	
Dry beans	6 (46.2%)	10 (76.9%)	0.157
Corn tortillas	4 (30.8%)	4 (30.8%)	
White tortillas	4 (30.8%)	3 (23.1%)	
Avg. number of foods available	2.77 (2.80)	3.31 (2.43)	0.131

*p*-values from McNemar’s exact tests or paired *t*-tests. McNemar’s exact tests only used where the availability of the item changed for at least one store. The following foods were not available anywhere pre- or post-: Black/African American: blackberries, blueberries, raspberries, chard, collard greens, kale, yams, rutabaga, peaches, beets, okra; East African: yams, corn, millet; Hispanic: acorn squash, rutabaga, cornmeal/masa. Asian stores (*n* = 2) are described in the manuscript text.

**Table 3 ijerph-18-05030-t003:** Percent of stores with any cultural food available in Minneapolis (versus St. Paul) by race/ethnicity neighborhood grouping pre- and 12-months post-ordinance enforcement.

Values are % any Cultural Food (SE)	*n*	Pre-	12-Months Post-	Change in %, β (95% CI)	*p*-Value Time	*p*-Value Time * City or Time *Ownership
All communities (Black/African American, East African, Asian or Hispanic)						
Model 1a						
Minneapolis	31	80.7 (7.9)	90.3 (6.4)	9.7 (−8.1–27.4)	0.279	0.683
St. Paul	29	65.5 (9.5)	69.0 (9.3)	3.4 (−21.2–28.1)	0.781	
Model 1b, adjusting for store ownership						
Minneapolis	31	80.8 (7.7)	86.8 (6.7)	6.1 (−11.0–23.1)	0.480	0.889
St. Paul	29	66.7 (9.4)	70.6 (9.2)	4.0 (−20.3–28.2)	0.743	
Model 2						
Minneapolis	31	81.0 (7.5)	88.9 (6.2)	8.0 (−10.0–25.9)	0.372	
Model 3						
Minneapolis						
Corporate	13	69.2 (13.4)	92.3 (8.4)	23.1 (−7.2–53.3)	0.129	0.213
Independent	18	88.9 (8.2)	88.9 (8.2)	0.0 (−21.4–21.4)	1.000	
Black/African American communities						
Model 1a						
Minneapolis	26	69.2 (10.0)	80.8 (8.8)	11.5 (−12.6–35.7)	0.339	0.366
St. Paul	13	76.9 (13.1)	69.2 (14.1)	−7.7 (−42.8–27.4)	0.660	
Model 1b, adjusting for store ownership						
Minneapolis	26	68.3 (9.4)	75.8 (8.5)	7.5 (−14.4–29.4)	0.492	0.523
St. Paul	13	75.4 (12.0)	70.5 (13.0)	−4.8 (−36.8–27.1)	0.761	
Model 2						
Minneapolis	26	67.6 (9.2)	74.3 (8.5)	6.7 (−14.9–28.3)	0.528	
Model 3 Minneapolis by ownership						

Notes. All models were generalized linear mixed models with a random intercept for store. Model 1a = Minneapolis and St. Paul stores, predictors: time, city, time*city; Model 1b = Minneapolis and St. Paul stores, predictors: time, city, time * city, ownership; Model 2 = Minneapolis stores only, predictors: time, ownership; Model 3 = Minneapolis stores only, predictors: time, ownership, time * ownership. * did not converge due to small sample size.

## Data Availability

The datasets used for the analyses presented in this manuscript are available online via the Data Repository for the University of Minnesota (https://conservancy.umn.edu/handle/11299/205351).

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
