# Peer review of "The Availability of Culturally Preferred Fruits, Vegetables and Whole Grains in Corner Stores and Non-Traditional Food Stores"

_ijerph, 2021, doi:10.3390/ijerph18095030_

Round 1

Reviewer 1 Report

First, this topic is of great importance as we understand how racial inequality and cultural implications have been left out of the food system for decades/centuries. I will say that I was given access to no tables to see the data and I have made the editors aware. I have based my review/questions off of what I was presented. 

How would stores be informed to align with the local ordinances? Were store owners/management provided the cultural equity toolkit? What exactly were retailers complying with and how was that enforced? Were there repercussions? 

Second to last sentence in first paragraph of materials and methods seems confusing and would benefit from rewording to clarify what stores were included in the study. 

Under Store Assessments, first sentence of the second paragraph - complied or compiled? Are all groups included in this study represented in the cultural equity toolkit? 

Within Analysis section, last sentence of the first paragraph - there are a few errors throughout the manuscript where multiple thoughts are trying to be expressed which results in a simple mistake like this. 

The way the analysis is framed it is one or more culturally relevant item. With few offering options - what is the benefit of having one or two cultural foods available? Is that beneficial? Does that mean anything? 

I also think seasonality matters. In the characterization of stores, did you account for or inquire into who suppliers were? Did they purchase local? Did they have access in their suppliers to these foods? Many would not be grown locally to MN. 

I think important considerations need to be made inr egards to the type of retailer. You hint at this in regards to corporate stores having more options available - have to consider small business owners making the decisions that will make them the most money. What about risk/benefit? Could modeling be done for small store owners to show benefit of increasing perishable foods when they make profit with unhealthy options? Could that be a future consideration? 

Is the time just 12 months between time 1 and time 4? I think you said this, but it got lost. Is that enough time for business to make substantial changes to supply chains? I really don't know. 

Finally, you mention 20% of a population of color living in census track and the idea of white flight. This is technically still the minority and it makes me think about purchasing power in those areas. Those who are spending the money are going to be driving market decisions. If there is still largely other populations in those areas, retailers will cater to the purchasing power and who is economically advantaged to stabilize them. Just something to think about.

I do want to close saying again I think this is incredibly timely and relevant. I'm excited to see this paper revised and the data files when they are available. I think the audience will find this informative and insightful and cause others to think about what is contributing to the established food systems in the communities they work and do their own research. Thank you for sharing your work. 

Reviewer 2 Report

General Comments: 

The manuscript is very well-written and adds to the knowledge of the effects of policy on food availability.  However, the appropriateness of the analyses was difficult to accurately assess since the tables were not included in the submission.

Specific comments:

  1. In the Methods, store ownership is mentioned as a possible confounder in change. Was store ownership considered in the regression models for overall availability between the cities of comparison?
  2. Paired t-tests were computed to compare the average number of culturally available food items. Was this change normally distributed?  Without the tables to review, normality was impossible to assess since no means and standard deviations were presented in the text of the manuscript.
  3. Availability appears to have been considered as a dichotomous variable (available/not available). However, the analysis methods section states that linear mixed models were run.  Logistic regression models would be more appropriate.    
  4. The ethnic breakdown of the stores in Minneapolis is presented, but not the breakdown for St. Paul and the comparison of the breakdown between cities is not presented or mentioned.  From the information provided, Asian communities were 26 overall and if only 2 were in Minneapolis, the other 24 must have been St. Paul which would most likely be a statistically significant difference.
  5. The set-up of the study was to compare Minneapolis to a control city (St. Paul).   However, very little attention is paid to those comparisons.  If the focus of this manuscript is on the intervention city and the changes seen, then the methods should be changed to reflect the intended focus.

Round 2

Reviewer 1 Report

Thank you for attention and thorough responses to my initial review. Seeing the tables helps a lot with understanding the analysis and results. I do feel that the additional clarity regarding analysis was beneficial, but it may be worth explicitly stating the normality of the data for comparisons. My final large consideration is what the paper gains from the comparison to St. Paul. In your hypothesis, you are mostly interested in assessing whether the ordinances made a difference in cultural offerings. Also, there is little discussion regarding the implications of the comparison. I simply ask you consider what this contributes and how the paper benefits from this analysis/comparison. Does it help or muddy the waters? Thanks again for providing a manuscript on such a relevant and important issue. I look forward to seeing more literature like this that leads to racial and health equity in the future. 

Author Response

Thank you for attention and thorough responses to my initial review. Seeing the tables helps a lot with understanding the analysis and results. Thank you

I do feel that the additional clarity regarding analysis was beneficial, but it may be worth explicitly stating the normality of the data for comparisons. Sentences were added (lines 171-175) in the methods stating that the data was not normally distributed and non-parametric tests showed the same results.

My final large consideration is what the paper gains from the comparison to St. Paul. In your hypothesis, you are mostly interested in assessing whether the ordinances made a difference in cultural offerings. Also, there is little discussion regarding the implications of the comparison. I simply ask you consider what this contributes and how the paper benefits from this analysis/comparison. Does it help or muddy the waters? I really appreciate this thoughtful point.  The value of having a comparison is that any changes we observed could be said to be due to the ordinance and not simply that there is a culture shift in what is available in small stores. I agree that the comparison is sort of 'dropped in' so I did a bit of reworking.

-at the end of the discussion (lines 87-89) a sentence was added to specify that the results will be compared to a control city to determine if the change is some larger industry process versus the implementation of the ordinance.

-In the discussion, I added more detail in lines 249-253 reflecting back on the hypothesis that change was due to ordinance and not industry trends (Comparison with St. Paul)

Thanks again for providing a manuscript on such a relevant and important issue. I look forward to seeing more literature like this that leads to racial and health equity in the future. Thank you

Reviewer 2 Report

The updated statistical methods section and minor contextual changes throughout the manuscript strengthens the methods and the discussion.  The additional information regarding the parent study was helpful for context. 

As for the statistical analyses, the normality of the number of foods available was not addressed in either the Response to Reviewers or the manuscript.  The standard deviations in Table 2 are approximately the same as the means and hence appropriateness of the two-sample t-test is questionable.

One minor comment on Table 1, the last 4 rows are missing notation regarding what is presented:  means (std) or n (%).

Author Response

Thank you for your review and request for clarification.

The updated statistical methods section and minor contextual changes throughout the manuscript strengthens the methods and the discussion.  The additional information regarding the parent study was helpful for context.  Thank you

As for the statistical analyses, the normality of the number of foods available was not addressed in either the Response to Reviewers or the manuscript.  The standard deviations in Table 2 are approximately the same as the means and hence appropriateness of the two-sample t-test is questionable. This is an excellent point. Sentences was added (line 171-175) reporting that the distribution was not normal and non-parametric tests resulted in the same conclusions.  

One minor comment on Table 1, the last 4 rows are missing notation regarding what is presented:  means (std) or n (%). Thank you - this was added.